H+ homeostasis; plasma membrane
H+-ATPase; protons; apoplast; cell surface.

**Corresponding authors:**Julien Gronnier;
Email: julien.gronnier@tum.de
Michael Palmgren;
Email: palmgren@plen.ku.dk

**Associate Editor:** Prof. Ingo Dreyer

# Molecular tipping points in plant cell surface H+ homeostasis and signalling

Kaltra Xhelilaj[1], Anja Thoe Fuglsang[2], Julien Gronnier[1,3] and Michael Palmgren[2]

[1]Center for Plant Molecular Biology (ZMBP), University of Tübingen, Tübingen, Germany; [2]Department of Plant and Environmental Sciences, University of Copenhagen, Frederiksberg C, Denmark; [3]Plant Cell Biology, School of Life Sciences, Technical University of Munich, Freising, Germany

## Abstract

Hydrogen, a deceptively simple element, plays crucial roles in regulating life on Earth. The concentration of hydrogen ions (H+) determines the pH of biological systems and dictates virtually all biochemical processes. The pH modulates the structure, physicochemical properties and function of most macromolecules. The plant cell surface is characterized by tremendous variations in apoplastic pH, serving as informative signals shaping plant development and its interaction with the environment. Here, we discuss the principles underlying cell surface H+ homeostasis, the molecular tipping points that regulate fast, controlled and informative changes in apoplastic pH, as well as open questions regarding the regulation of plasma membrane H+-ATPases.

## 1. Cell surface H+ homeostasis

Establishing an H+ gradient across biological membranes and between tissues is essential for cellular processes, influencing plant development, nutrition and immunity. At the plant cell surface, striking differences in pH are observed on both sides of the plasma membrane (PM). The cytosolic pH is tightly regulated at constant alkaline values (pH 7.3–8), while the apoplastic pH is more acidic with important pH value fluctuations (pH 4–6.3) (Felle et al., 2005; Geilfus, 2017; Martinière et al., 2013; Shen et al., 2013). Sustaining this steep gradient demands a multilayered regulatory system operating simultaneously. A first layer of regulation is based on biochemical and chemical buffering capacities that stabilize pH. For example, H+ consumption during reactions of malate or glutamate decarboxylation buffers cytosolic pH (Reguera et al., 2015; Gerendá & Schurr, 1999). Furthermore, P-type PM H+-ATPases function as H+ extrusion pumps that are activated from the cytoplasmic side following acidification of the cytoplasm (Regenberg et al., 1995). Thus, these pumps act as molecular 'pH-stats', safekeeping defined cytosolic pH values (Contador-Álvarez et al., 2025; Regenberg et al., 1995)

By contrast, the apoplast buffering capacity is tenfold lower, making the apoplast prone to rapid pH changes (Felle & Hanstein, 2002; Hanstein & Felle, 1999; Oja et al., 1999). H+ influx and efflux across PM play an important role in fine-tuning pH modifications (Figure 1). The PM H+-ATPases actively pump H+ into the apoplast, generating the proton motive force (PMF), a critical gradient that energizes secondary active transport systems (Palmgren, 2001). The generation of the PMF is hypothesized to have driven the evolution of symporters and antiporters for molecules and ions, which in turn often influence pH as well (Nelson, 1994). Examples of molecule/H+ symporters include NITRATE TRANSPORTER1 and AUXIN-RESISTANT1 (AUX1)/LIKE-AUX1 importers (Dindas et al., 2018). A prominent example of a molecule/H+ antiporter is the Na+/H+ transporter Salt Overly Sensitive1 (SOS1), which has been proposed to adjust pH in the short term as an alternative and in addition to the PM H+-ATPases (Felle, 1989). In addition to active transport and PMF-driven fluxes, passive H+ diffusion across the PM lipid bilayer can also influence pH dynamics, particularly under stress conditions that alter PM biophysical properties (Dhindsa et al., 1981; Larkindale & Huang, 2004; Willing & Leopold, 1983).

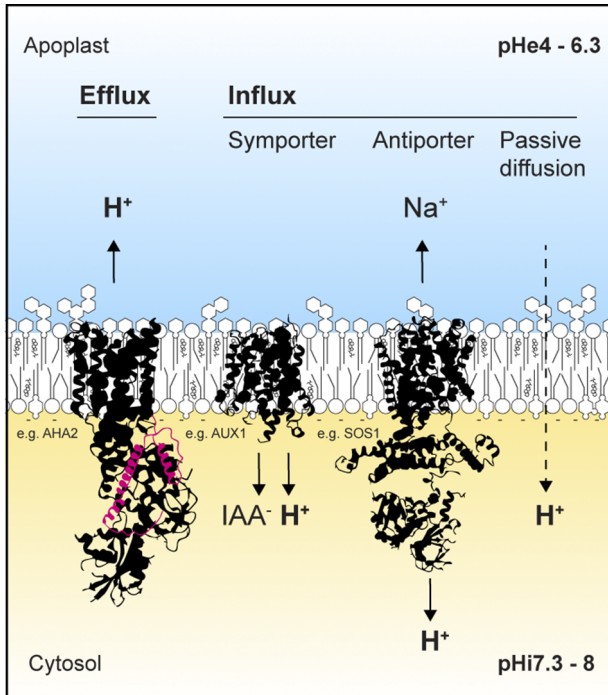

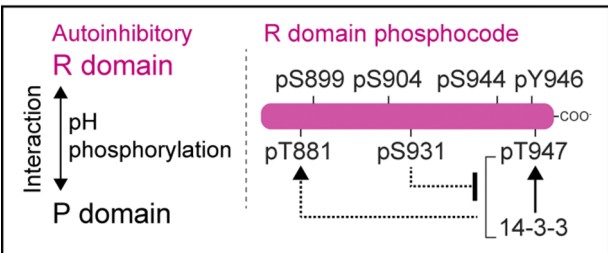

**Figure 1.** H$^+$-transport and diffusion across the plant plasma membrane. H$^+$ can exit the cell (efflux) or enter the cell (influx). The plasma membrane H$^+$-ATPases drive H$^+$ efflux against an H$^+$ gradient, generating the proton motive force. Influxes of H$^+$ can be mediated by symporters, antiporters or by passive diffusion. Schematic representations of AHA2, AUX1 and SOS1 structures are depicted as examples. The regulatory R-domain of AHA2 is depicted in pink. It is noteworthy that the stoichiometry of H$^+$/IAA- cotransport by AUX1 is still a matter of debate (Geisler & Dreyer, 2024). Bottom: PM H$^+$-ATPase autoinhibition is driven by the R-domain and its interaction with the P-domain. This interaction is sensitive to pH, and modulated by phosphorylation, and phosphorylation-dependent binding of 14-3-3, conceptualized as an R-domain phosphocode.

## 2. Multilayered regulation of cell surface H$^+$ in development and immunity

H$^+$ homeostasis is inherently regulated in virtually all aspects of a plant's life. According to the acid growth theory, described more than 50 years ago, the activation of PM H$^+$-ATPases acidifies pHe, which promotes cell wall loosening and thus cell elongation (Hager et al., 1971; Rayle & Cleland, 1970, 1980). In contrast, extracellular pH (pHe) alkalinization promotes cell wall stiffening and slows growth (Geilfus, 2017). In recent years, our understanding of pHe regulation has expanded beyond the classic acid growth theory. Indeed, pH is now recognized as a key regulatory factor in many cellular processes, with broader and more complex roles than previously envisioned. For example, fluctuations in apoplastic pH strongly influence the reactivity of extracellular H$_2$O$_2$ in oxidizing thiols of PM proteins (Zhou et al., 2025). Another example is the pH variation across root layers and how alkalinization of

protophloem sieve elements leads to different signalling responses and subsequently influences the differentiation of protophloem sieve elements and thus root growth (Diaz-Ardila et al., 2023; Diaz-Ardila & Hardtke, 2025). Various aspects of H$^+$ homeostasis have been previously covered for different aspects of plant physiology (Falhof et al., 2016), abiotic stress (Li & Yang, 2023) and hormone signalling (Miao et al., 2021), as well as in tissue (Gámez-Arjona et al., 2022) and cell-type-specific contexts (Stéger et al., 2022). Hereafter, we focus on the stimuli-dependent regulation of the commonly targeted PM H$^+$-ATPases and the regulation of H$^+$ fluxes by auxin signalling and during plant immune signalling. These mechanisms evolved early in plant evolution and have been proposed to represent an adaptation to the water-to-land transition of plants (Stéger et al., 2022; Zeng et al., 2024).

### 2.1. PM H$^+$-ATPases and pH-stats under tight control

PM H$^+$-ATPases belong to the superfamily of P-type ATPases and consist of a single polypeptide chain folding into a multi-domain structure (Palmgren, 2023). The core architecture of PM H$^+$-ATPases follows the typical P-type ATPase structure, consisting of four domains: N (nucleotide-binding), P (phosphorylation), A (actuator) and M (membrane) domains that comprise 10 transmembrane domains (Kühlbrandt, 2004; Pedersen et al., 2007). In yeast and plants, PM H$^+$ ATPases carry sequence extensions at both their N- and C-termini, which are implicated in regulation and autoinhibition (Ekberg et al., 2010; Falhof et al., 2016; Kühlbrandt et al., 2002; Palmgren et al., 1991). In the green lineage, PM H$^+$-ATPases C-termini extend further into an R (regulatory) domain that includes multiple phosphorylation sites (Rudashevskaya et al., 2012). Cross-linking experiments showed that the R-domain can extensively cross-link with the A-, N- and P-domains, as well as the H$^+$-binding site, suggesting that there is an intramolecular R-domain-mediated autoinhibition (Blackburn et al., 2025; Nguyen et al., 2020).

PM H$^+$-ATPases are normally autoinhibited at cytosolic pH but become strongly activated if the cytosol acidifies (Luo et al., 1999; Palmgren & Christensen, 1994; Regenberg et al., 1995). The autoinhibition of the PM H$^+$-ATPase can be overruled during signalling. Indeed, numerous signalling pathways in development, reproduction and immunity target the activity of the PM H$^+$-ATPases and converge in modulating the phosphorylation level of the R-domain (Falhof et al., 2016; Fuglsang & Palmgren, 2021; Haruta et al., 2015). These studies unveiled a common regulatory scheme, an R-domain phosphocode, from which the contribution of individual residues, their hierarchy and the sequentiality of phosphorylation events emerge. The phosphorylation of the R-domain can be translated as positive or repressive marks, promoting or inhibiting PM H$^+$-ATPase activity (Figure 1).

Of prominent importance is the phosphorylation of the conserved penultimate residue (a threonine; T947 in the *Arabidopsis* AUTOINHIBITED H$^+$-ATPase 2; AHA2), which is commonly linked to AHA activation (Fuglsang et al., 1999; Svennelid et al., 1999). Phosphorylation of this residue creates a binding site for 14-3-3 proteins, which, upon binding, releases C-terminal autoinhibition (Fuglsang et al., 1999; Jahn et al., 1997). Another positive mark is the phosphorylation at T881 (AHA2) (Fuglsang et al., 2014; Hayashi et al., 2024; Wang et al., 2022). Interestingly, time-resolved quantitative phosphoproteomics and genetic experiments indicate that in response to blue light, phosphorylation of the penultimate residue precedes and conditions the phosphorylation of T881 (pT881) in AHA1 (Hayashi et al., 2024). This functional

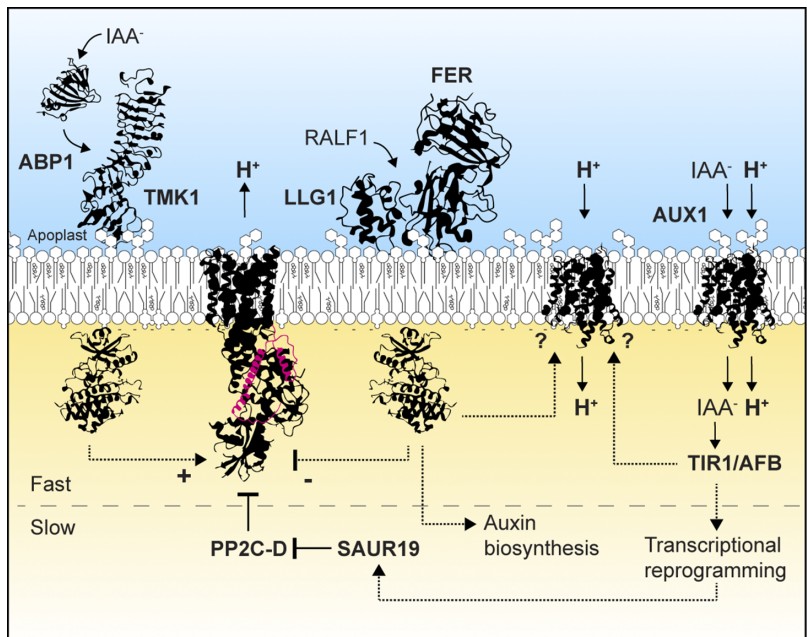

**Figure 2.** Molecular circuitry of auxin-mediated regulation of extracellular pH. Auxin signalling is categorized into two main branches: fast non-transcriptional cellular responses and slower transcriptional responses that converge in the regulation of pHe. ABP1, AUXIN-BINDING PROTEIN 1; AFB, AUXIN-SIGNALLING F-BOX; AUX1, AUXIN-RESISTANT 1; FER, FERONIA; IAA, Indole-3-Acetic Acid; LLG1, LORELEI-LIKE GPI-ANCHOR PROTEIN 1; PP2C-D, type 2C protein phosphatase clade D; RALF1, RAPID ALKALINIZATION FACTOR1; SAUR19, SMALL AUXIN Up-RNA 19; TIR1, TRANSPORT INHIBITOR RESPONSE 1; TMK1, TRANSMEMBRANE KINASE 1. It is noteworthy that the stoichiometry of $H^+$/IAA- cotransport by AUX1 is still a matter of debate (Geisler & Dreyer, 2024).

relationship appears unilateral as pT881 is dispensable for the phosphorylation of T947 (pT947) (Fuglsang et al., 2014). In contrast to the pT947 and pT881, phosphorylation at S931 (pS931) (AHA2) corresponds to a repressive mark linked to AHA inhibition. pS931 prevents 14-3-3 binding to the R-domain independently of the phosphorylation status of T947 (Duby et al., 2009; Fuglsang et al., 2007). Similarly, the phosphorylation of S899 (pS899) (AHA2) has been linked with inhibition of PM $H^+$-ATPases (Haruta et al., 2014; Zhu et al., 2024). How pS899 inhibits AHA activity, and its relationship with other residues, is, however, unknown. Similarly, the function of other residues identified in phosphoproteomic experiments remains unknown (Fuglsang & Palmgren, 2021; Rudashevskaya et al., 2012).

## 2.2. Molecular circuitry of auxin-mediated regulation of pHe and growth

The phytohormone auxin controls many processes in plants, including cell expansion, cell division and cell differentiation (Du et al., 2020; Enders & Strader, 2015). Auxin is perceived at the cell surface via the auxin-binding proteins (ABPs) and their plasma membrane partner, the transmembrane kinases (TMKs) (Friml et al., 2022), and intracellularly by TIR/AFB–Aux/IAA receptor–coreceptor modules (Dharmasiri et al., 2005; Kepinski & Leyser, 2005; Tan et al., 2007). Auxin perception induces two main signalling branches: (i) a fast cellular auxin response that includes PM depolarization, changes in extracellular pH (pHe), $Ca^{2+}$ influx and root growth inhibition (Ayling et al., 1994; Barbez et al., 2017; Monshausen et al., 2011); and (ii) a slower nuclear auxin pathway regulating a broad transcriptional response (Leyser, 2018; Weijers & Wagner, 2016) (Figure 2). The effect of auxin on pHe is tissue-, concentration- and time-dependent.

In *Arabidopsis hypocotyls*, transcriptional and non-transcriptional auxin responses converge towards the activation of PM $H^+$-ATPases, the acidification of the apoplast and the promotion of cellular expansion (Fendrych et al., 2016; Lin et al., 2021; Ren et al., 2018; Spartz et al., 2014). Upon auxin perception by ABP1-TMK1, TMK1 phosphorylates and activates the PM $H^+$-ATPase AHA1 within seconds (Li et al., 2021; Lin et al., 2021), providing a direct molecular link between auxin perception and acidification of the apoplast. Among the transcriptional targets regulated by auxin and its intracellular receptor is SMALL AUXIN Up-RNA 19 (SAUR19). SAUR19 binds to and inhibits the TYPE 2C PROTEIN PHOSPHATASES, which normally dephosphorylates and inhibits the activity of the PM $H^+$-ATPases. Thereby, SAUR19 promotes PM $H^+$-ATPase activity, acidification of the apoplast and cell expansion (Fendrych et al., 2016; Ren et al., 2018; Spartz et al., 2014). Recently, pHe was proposed to be a key switch in auxin-induced hypocotyl elongation. Auxin-driven acidification promotes elongation until an optimal pHe threshold is reached (Wang et al., 2025). Beyond this optimal pHe, elongation ceases due to a negative feedback loop, which results in a biphasic hypocotyl elongation and suggests a 'gas then break' mechanism for the fine-tuning of hypocotyl growth. Additional variables might be involved in this mechanism, such as light signalling, which antagonizes the influence of auxin on both pHe and cell elongation (Wang et al., 2025). In *Arabidopsis* root, endogenous auxin signalling is required for apoplast acidification, cellular elongation (Barbez et al., 2017; Li et al., 2021) and guide gravitropic and hydotropic root navigation in the soil environment.

## 2.3. Auxin-induced extracellular alkalinization in roots

Applied exogenously, nanomolar auxin concentrations trigger fast and reversible inhibition of root growth, which is linked to an

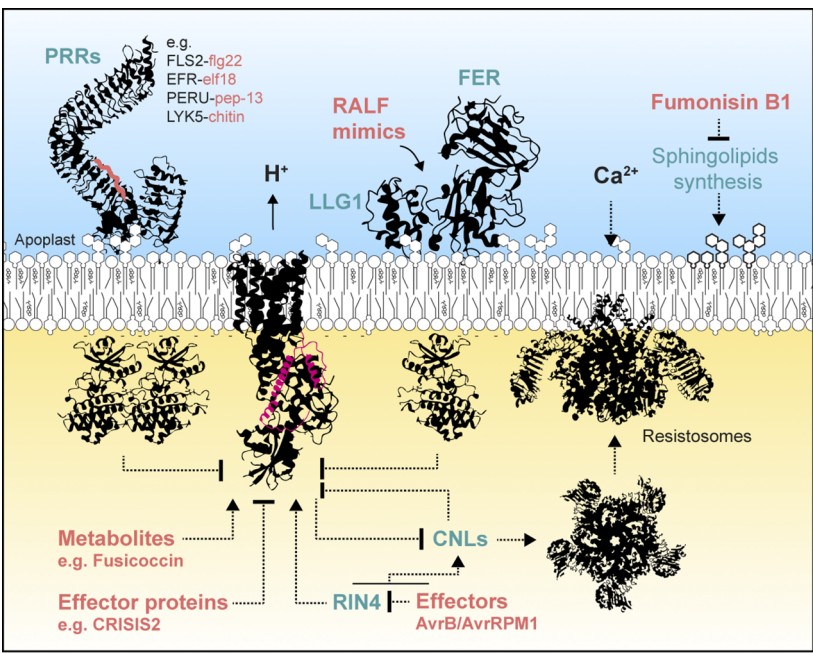

**Figure 3.** Plasma membrane H$^+$-ATPases are central molecular nodes in immunity. Both pattern-triggered immunity and effector-triggered immunity converge on the regulation of PM H$^+$-ATPases. Pathogens evolved an array of strategies to inhibit or activate PM H$^+$-ATPases. The plant molecular components are named in green, and pathogen-derived molecules are named in red. CNLs, coil-coiled nucleotide-binding leucine-rich repeat receptors; FER, FERONIA; LLG1, LORELEI-LIKE GPI-ANCHOR PROTEIN 1; PRR, pattern recognition receptor; RALF, RAPID ALKALINIZATION FACTOR; RIN4, RPM1-INTERACTING PROTEIN4.

alkalinization of the apoplast (Barbez et al., 2017; Li et al., 2021). This suggests that auxin promotes an inward H$^+$ flow, evoking alkalinization of the apoplast and accompanying depolarization of the PM. The auxin-triggered alkalinization relies on the IAA/H$^+$ symporter AUX1, the non-transcriptional action of the intracellular auxin receptors and a CNGC14-mediated Ca$^{2+}$ influx (Dindas et al., 2018; Li et al., 2021; Serre et al., 2021, 2023). AUX1 IAA/H$^+$ symporter activity is expected to directly contribute to H$^+$ influx. However, despite the fact that loss of AUX1 abolishes membrane depolarization in root hairs (Dindas et al., 2018), AUX1 has a minor contribution to auxin-induced membrane depolarization at the root tip (Serre et al., 2021) and the intracellular injection of auxin is sufficient to induce depolarization (Dindas et al., 2018). Furthermore, comparative analysis of the speed of H$^+$ influx and auxin uptake indicates that auxin transport itself cannot fully explain the H$^+$ influx, which would be predominantly executed by an unknown auxin-stimulated H$^+$ permeable channel (Li et al., 2021). Many signalling pathways crosstalk with auxin signalling and converge in the regulation of pHe and H$^+$-ATPase. For instance, the cell surface receptor modules of both brassinosteroid and auxin directly phosphorylate and activate PM H$^+$-ATPases to promote wall acidification and cell expansion (Miao et al., 2022). Conversely, in the root, the perception of RAPID ALKALINIZATION FACTOR peptides (RALFs) and auxin converge in pHe alkalinization (Abarca et al., 2021; Gjetting et al., 2020; Li et al., 2022; Morato do Canto et al., 2014). The effect of RALF1 on apoplastic pH is proposed to be linked to the inhibition of the PM H$^+$-ATPase (Haruta et al., 2014), albeit genetic experiments rather suggest that the activation of an unknown influx carrier is responsible for RALF1-triggered net H$^+$ influx (Li et al., 2022). Furthermore, RALF1 perception promotes auxin biosynthesis and signalling, thereby sustaining its effect on growth (Li et al., 2022).

As in the hypocotyl, TMK1-mediated activation of PM H$^+$-ATPases occurs in response to auxin treatment and counterbalances the dominating alkalinization (Li et al., 2021). This intuitively argues against PM H$^+$-ATPases playing a role in the auxin-induced alkalinization, as it is explained by a net H$^+$ influx, the mechanism of which remains unknown (Li et al., 2021, 2022; Serre et al., 2023). However, the PM H$^+$-ATPase can also attain an uncoupled state that is either leaky to H$^+$ or exhibits slippage (ATP hydrolysis without accompanying H$^+$ transport) (Baunsgaard et al., 1996; Morsomme et al., 1996; Pedersen et al., 2018). Similarly, the animal Na$^+$/K$^+$-ATPase, a related P-type ATPase, can be converted into a channel protein that causes the direction of Na$^+$ transport to be downhill (Reyes & Gadsby, 2006; Scheiner-Bobis & Schneider, 1997). Whether the uncoupling of the PM H$^+$-ATPase contributes to the initial rapid extracellular alkalinization in roots remains to be investigated.

## 2.4. PM H$^+$-ATPases as convergent nodes linking pattern- and effector-triggered immunities

Changes in pHe have been repetitively observed in plant–microbe interactions (Elmore & Coaker, 2011; Felle et al., 2004; Kesten et al., 2019; Vera-Estrella et al., 1994). Plants perceive microbes via cell surface and intracellular immune receptors involved in pattern-triggered immunity (PTI) and effector-triggered immunity (ETI) (Ngou et al., 2022). At the cell surface, pattern recognition receptors (PPRs) sense microbe- or self-derived molecular patterns and initiate an array of molecular events culminating in PTI (DeFalco & Zipfel, 2021). Together with the production of reactive oxygen species and influx of Ca$^{2+}$, the rapid alkalinization of the apoplast corresponds to a rapid molecular hallmark of PPR signalling (Boller & Felix, 2009) (Figure 3). Indeed, rapid changes

in apoplastic pH have been observed in various plant species irrespective of the biochemical nature and the microbial origin of the molecular pattern perceived. For instance, perception of the fungal cell wall component chitin, the oomycete-derived peptide pep-13, the bacterial flagellin epitope flg22 and the endogenous danger-associated molecular patterns pep1 all lead to alkalinization of pHe (Felix et al., 1993; Liu et al., 2022; Nürnberger et al., 1994; Yamaguchi et al., 2006). Untargeted phosphoproteomic studies indicate that it may be mediated by the inhibition of PM H$^+$ ATPases, as the perception of flg22 leads to a decrease in the phosphorylation of residues known to promote PM H$^+$-ATPase activity (e.g., AHA2 T947; Benschop et al., 2007; Nühse et al., 2007), and the perception of flg22 and oligogalacturonides promotes the phosphorylation of a residue linked to the inhibition of the PM H$^+$-ATPase (AHA2 S899; Benschop et al., 2007; Mattei et al., 2016; Nühse et al., 2007).

Akin to their cell surface counterpart, intercellular immune receptors have been shown to modulate pHe. Indeed, in *Nicotiana benthamiana* several members of a subgroup of intracellular nucleotide-binding leucine-rich repeat receptors (NLRs), namely the plasma membrane-localized coiled-coil NLRs (CNLs) (Saile et al., 2021), have been reported to inhibit PM H$^+$ ATPase activity, resulting in apoplastic alkalization (Lee et al., 2022). Interestingly, genetic and pharmacologic approaches indicate that inhibition and hyper-activation of PM H$^+$ ATPase activity affect CNL function (Lee et al., 2022). Upon activation, CNLs form a PM-localized Ca$^{2+}$-permeable wheel-like oligomeric structures mediating a Ca$^{2+}$ influx for ETI signalling (Wang et al., 2023). Whether balanced H$^+$-ATPase activity is required for the activation of CNLs or CNL-mediated execution of cell death is currently unknown. The two CNLs RPS2 and RPM1 guard the plant protein RPM1-INTERACTING PROTEIN4 (RIN4) and activate ETI upon sensing effector-mediated RIN4 modification (Axtell & Staskawicz, 2003; Mackey et al., 2002). Interestingly, RIN4 was shown to directly associate and promote the activity of H$^+$ ATPase (Liu et al., 2009). Further, during infection, in the presence of the *Pseudomonas syringae* effector AvrB, RPM1-INDUCED PROTEIN KINASE phosphorylates RIN4 at T166, thereby promoting its association with AHA1 (Lee et al., 2015). As flg22 perception inhibits RIN4 phosphorylation at this residue, the inhibition of PM H$^+$-ATPase upon flg22-triggered signalling may, in part, be mediated by the regulation of RIN4 (Lee et al., 2015). Altogether, these studies place PM H$^+$-ATPases as a central node linking the two main branches of the plant immune system, which may contribute to their functional relationship (Feehan et al., 2023; Ngou et al., 2021; Wang et al., 2023).

Interestingly, PM H$^+$-ATPases are among the limited number of proteins differentially phosphorylated upon both ETI and PTI (Kadota et al., 2019). Given the apparent importance of PM H$^+$-ATPase in immune signalling, it is not surprising that it is a common target of pathogens that promote disease. Indeed, pathogens have been shown to utilize metabolites and protein effectors to manipulate PM H$^+$-ATPase (Figure 3). A prominent example is fusicoccin, a diterpene glucoside produced by the pathogenic fungus *Fusicoccum amygdale*, which irreversibly stabilizes the interaction between the 14-3-3 and the C-terminal regulatory domain of PM H$^+$-ATPases (Baunsgaard et al., 1998; Jahn et al., 1997), thereby relieving auto-inhibition and leading to constitutive H$^+$ pumping. As stomatal pores open in response to activation of H$^+$ pumping by the PM H$^+$-ATPase (Cha et al., 2024; Hayashi et al., 2024; Kinoshita & Shimazaki, 1999), this provides an entry point for the pathogen. Fusicoccin has been

shown to cause continuous stomatal opening, wilting and necrosis of leaves (Elmore & Coaker, 2011). By contrast, tenuazonic acid produced by fungi, such as *Stemphylium loti*, inhibits PM H$^+$-ATPase activity (Bjørk et al., 2020; Havshøi et al., 2024). Several protein effectors have been shown to target PM H$^+$-ATPase activity as well. For instance, the *Phytophthora infestans* RxLR effector PITG06478 hijacks 14-3-3 proteins to suppress PM H$^+$-ATPase activity (Seo et al., 2023), and the *Phytophthora capsici* effector CRISIS2 associates with and inhibits PM H$^+$-ATPases to promote disease (Seo et al., 2023). By disrupting PM H$^+$-ATPase function, these effectors are proposed to facilitate successful microbial proliferation. However, the underlying molecular mechanisms remain unknown. In addition, pathogens and parasites utilize plant mimicking RALF peptides to subvert PM H$^+$-ATPases to their advantage (Masachis et al., 2016; Wang et al., 2024; Wood et al., 2020; Zhang et al., 2020). Still, many key questions remain to be answered. For instance, how signalling is relayed from activated immune complexes to the PM H$^+$-ATPase is currently unknown. Further, while the regulation of PM H$^+$-ATPase has been functionally linked with the regulation of stomatal opening, thereby limiting the entry of bacteria inside plant tissues (Liu et al., 2009; Melotto et al., 2006), the role of pHe in immune signalling and plant immunity in other cell types remains unclear.

## 2.5. Feedback loops in pH sensing and signalling

How the plant cell senses pHe has long been enigmatic (Tsai & Schmidt, 2021). Accumulating evidence from structural biology studies emphasizes the implication of cell surface ligand–receptor modules in pH sensing and signalling (Xu & Yu, 2023). For instance, the brassinosteroid-induced association between the main brassinosteroid receptor Brassinosteroid-Insensitive1 (BRI1) and its co-receptor BRI1-Associated Receptor Kinase1 (BAK1) is promoted by a relatively acidic pH *in vitro* (Sun et al., 2013). On the contrary, relatively alkaline pH promotes the binding of the peptide pep1 to its corresponding receptors PEPR1 and PEPR2 (Liu et al., 2022; Tang et al., 2015). This is explained by reversible (de-)protonation of amino acid residues and such direct interaction with H$^+$, which occurs within the ligand or at the interface between the receptor and co-receptors (Xu & Yu, 2023). These observations recently found a biological echo in the balance between plant immunity and growth. Indeed, pep1-triggered immune signalling alkalinizes pHe in the root apical meristem, thereby inhibiting the perception of root meristem growth factor 1 by its corresponding receptor, and ultimately growth (Liu et al., 2022).

Several of these cell surface receptors have been shown to associate and directly phosphorylate AHAs. For instance, as discussed above, auxin perception induces TMK1–AHA1 interaction and AHA1 phosphorylation by TMK1 (Li et al., 2021). Similarly, AHA1 associates with and is activated by BRI1 (Caesar et al., 2011). Conversely, the receptor kinase Qiān Shǒu Kinase1 has been proposed to directly phosphorylate and inhibit AHA2 to mediate a low nitrate response (Zhu et al., 2024). How these stimuli-dependent associations are regulated is currently unknown. Finally, it is interesting to note that cell surface signalling pathways that are activated at relatively low pHe have been shown to promote AHA activity (Großeholz et al., 2022; Sun et al., 2013). Conversely, cell surface pathways activated at relatively high pHe have been shown to inhibit AHA activity (Liu et al., 2022; Tang et al., 2015) (Figure 4). This suggests that cell surface pathways operate positive feed-forward loops to optimize selfishly their own signalling to the

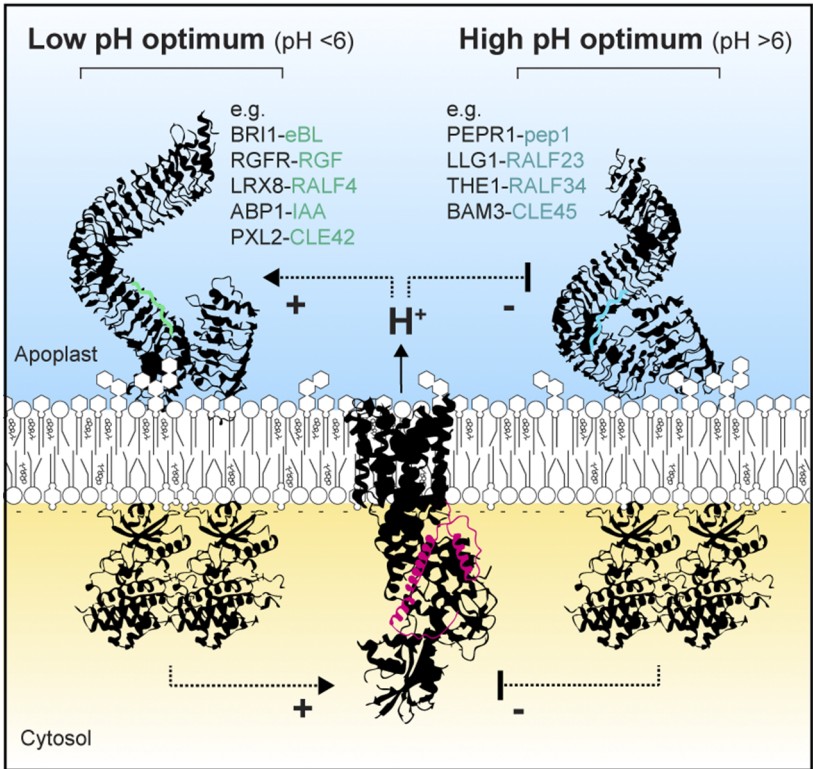

**Figure 4.** Feedback loops in pH sensing and signalling. Through pH-sensitive ligand–receptor and ligand-induced receptor interactions, cell surface receptors and their corresponding ligands play preponderant role in pHe sensing. They can be categorized into two classes having relatively low or high pH optimum and signalling back to the PM $H^+$-ATPases through feedback loops.

detriment of others. How this is balanced and regulated as part of the plant developmental programme is unknown.

## 3. Architecture and nano-environment of PM $H^+$-ATPases complexes

### 3.1. PM $H^+$-ATPase oligomerization, strength in numbers?

Isolated monomers of PM $H^+$-ATPase lacking the C-terminal autoinhibitory domain are functional *in vitro* in membrane nanodiscs (Justesen et al., 2013), suggesting that there is no strict additional molecular requirement for the $H^+$ translocation capability. In plants, however, PM $H^+$-ATPases exist in various oligomerization states (Kanczewska et al., 2005). Consistent with higher-order assemblies of PM $H^+$-ATPase, cryogenic electron microscopic (cryo-EM) studies of plant and fungal PM $H^+$-ATPases resolved the organization and structure of hexamers (Cyrklaff et al., 1995; Heit et al., 2021; Ottmann et al., 2007; Zhao et al., 2021). In yeast, PM $H^+$-ATPases appear to be produced and transported from the endoplasmic reticulum as hexamers (Lee et al., 2002). In plants, PM $H^+$-ATPase seems to be predominantly found in the form of dimers (Kanczewska et al., 2005). The strong and irreversible activation induced by fusicoccin treatment leads to the formation of higher-order oligomers (Kanczewska et al., 2005) and phosphorylation-dependent binding of 14-3-3 dimers to dimeric $H^+$-ATPases is proposed to lead to the assembly of $H^+$-ATPase hexamers (Ottmann et al., 2007). Cross-linking experiments further suggest 'head-to-tail' interactions between the R-domains of neighbouring monomers (Nguyen et al., 2020). However, whether PM $H^+$-ATPase oligomerization occurs during plant signalling and what could be the associated functional

consequences remain largely unclear. It is tempting to speculate that the formation of such oligomers could drive important and localized changes in pHe (Figure 5).

PM $H^+$-ATPases form a multigenic protein family in plants, with 11 isoforms encoded in the genome of *Arabidopsis thaliana* Col-0, for instance (Palmgren, 2001). Whether the assembly of dimers and higher-order oligomers involves heteromerization of AHA isoforms is currently unknown. Co-immunoprecipitation experiments indicate that different isoforms can be found in proximity (Rodrigues et al., 2014) and suggest the formation of heteromeric complexes. Given the differences in $H^+$ translocation efficiency and pH sensitivity among AHA isoforms (Hoffmann et al., 2019; 2020), the potential formation of hetero-oligomers may serve for finely tuned pHe variation in cell-type and stimuli-dependent conditions.

### 3.2. A functional $H^+$-ATPase paralipidome?

The sensitivity of $H^+$-ATPases to lipid moieties has long been described (Palmgren et al., 1988). For instance, phospholipid species, such as lyso-phosphatidylcholine (lyso-PC) (Palmgren & Sommarin, 1989) and phosphatidylserine (PS) (Paweletz et al., 2023), have been shown to stimulate $H^+$ pumping by the PM $H^+$-ATPase. Membrane lipids can serve simultaneously as solvents and regulatory co-factors for membrane proteins, constituting a so-called paralipidome (Leventhal & Lyman, 2022). Further, PM lipids and proteins are dynamically organized into numerous membrane nano-environments termed nanodomains (Jaillais et al., 2024). In the yeast *Saccharomyces cerevisiae*, the PM $H^+$-ATPase1 (PMA1) is confined to subregions of the PM forming a reticulated

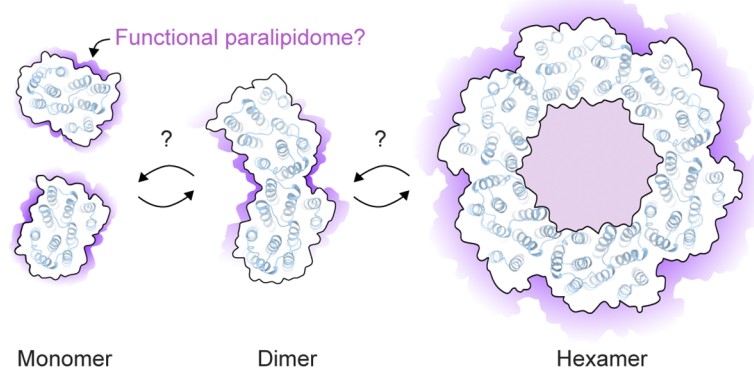

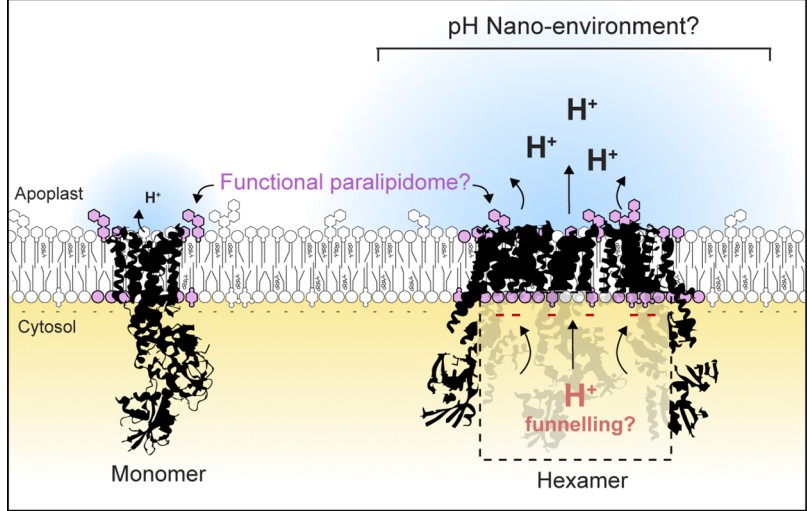

**Figure 5.** Unknown molecular assembly of plant plasma membrane H⁺-ATPases and surrounding lipids. Top view of schematic representation of the PM H⁺-ATPase transmembrane alpha helices. While the cryo-EM studies of plant, yeast and fungus highlight PM H⁺-ATPase homo-hexamers, the potential dynamic assembly of oligomers *in vivo* and their potential function remain largely unknown in plants. The light pink membrane areas depict a putative functional paralipidome in which specific lipids are hypothesized to fine-tune PM H⁺-ATPase activity.

Bottom: Side view of PM H⁺-ATPase monomer and hexamer representing a putative phosphatidylserine-driven H⁺ funnelling and the generation of a pHe nano-environment.

mesh-like network that is spatially distinct from the membrane compartments formed by the amino acid transporter Canavanine-resistance1 (Can1) (Malínská et al., 2003; Spira et al., 2012). A comparative lipidomic analysis of PMA1- and Can1-containing membrane nano-environments indicates that sphingolipids and PS are enriched in the vicinity of PMA1 (van't Klooster et al., 2020).

Interestingly, very long chain-containing sphingolipids are required for the oligomerization and PM localization of yeast PMA1 (Lee et al., 2002; Gaigg et al., 2006; Wang & Chang, 2002) and cryo-EM analysis showed that the PMA1 hexamer encircles lipids of the outer leaflet, forming a liquid-crystalline patch of membrane that is presumably composed of sphingolipids (Zhao et al., 2021). Plant and yeast PM H⁺-ATPases co-purify with detergent-resistant membrane biochemical fractions (Bagnat et al., 2001; Mongrand et al., 2004) indicating that they present similar membrane biochemical properties, which may be linked to sphingolipids. Plant sphingolipids play preponderant roles in regulating PM structure and function (Gronnier et al., 2016; Mamode Cassim et al., 2019). Interestingly, the inhibition of ceramide synthase involved in sphingolipid synthesis by the fungal toxin fumonisin B1 (Wang et al., 1991; Zhang et al., 2024) affects PM H⁺-ATPase activity in maize embryos (Gutiérrez-Nájera et al., 2005; Gutiérrez-Nájera et al., 2020), suggesting that a functional

interplay between PM H⁺-ATPases and sphingolipids exists in plants as well (Figures 4 and 5).

Molecular dynamic simulations of *Neurospora crassa* PMA1 oligomers predict that PS binds to PMA1 and accumulates at the interface between monomers (Heit et al., 2021). Similarly, molecular dynamic simulations of AHA2 indicate preferential association with PS (Paweletz et al., 2023). PS has been suggested to promote the assembly of PM H⁺-ATPase hexamers (Heit et al., 2021). Further, the polar head of PS being electronegative, its accumulation within the vicinity of PMA1 hexamer may serve to attract and funnel H⁺ to foster H⁺ translocation by the PM H⁺-ATPases (Heit et al., 2021) (Figure 5). Corroborating these predictions, PS is particularly efficient in promoting the activity of *Arabidopsis* AHA2 in proteoliposomes (Paweletz et al., 2023). However, the exact molecular mechanism for the regulation of H⁺-ATPases activity by PS remains to be determined. In comparison with yeast PMA1, which has served as a model protein to study PM organization (Malinsky et al., 2013), much less is known about the nanoscale organization of its plant homologues. In *Arabidopsis*, single-particle tracking photoactivated localization microscopy experiments showed that AHA2 diffuses slowly within the PM, and that osmotic stress enhances AHA2 diffusion (Martinière et al., 2019). Long-term single-molecule imaging indicates that slow

diffusing AHA2 is transiently spatially confined in membrane nano-environments (von Arx et al., 2024). The molecular bases of these events and their functional relevance are, however, unknown. It would be of particular interest to investigate the potential dynamic formation of PM H$^+$-ATPase complexes and potential nanoclusters in living plant cells.

**Open peer review.** To view the open peer review materials for this article, please visit http://doi.org/10.1017/qpb.2025.10002.

**Data availability statement.** The data that support the findings of this study are openly available in the cited references.

**Author contributions.** Kaltra Xhelilaj and Julien Gronnier wrote the first draft of the review. Julien Gronnier made the figures. All authors contributed to the final version.

**Funding statement.** This work was supported by the Novo Nordisk Foundation (Michael Palmgren, NovoCrops NNF19OC005658), the Innovation Fund Denmark (Michael Palmgren, DEEPROOTS and PERENNIAL) and the Deutsche Forschungsgemeinschaft (Julien Gronnier, A08-SFB1101 and B01-TRR356).

**Competing interest.** The authors declare none.

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
