## [Reviewer Report]

This is a competent and authoritative review that covers extracellular pH homeostasis plants from the perspective of the molecular machinery that moves protons across the plasma membrane and the way in which the relevant proteins are regulated. The review covers contemporary literature in the area effectively, although it is rather narrow in scope.

Plants grow in a wide variety of acidic and alkaline environments. An ecologist looking at this paper might justifiably expect a discussion of how acid and alkalophilic ecotypes might evolve in the context of a soil proton activity that varies across six orders of magnitude. If the title is to be retained, some element of wider environmental context must be given.

Also lacking in the review is a perspective on the potentially conflicting roles of plasma membrane proton transport systems. Molecular, but not cellular, detail is given. For example, a plant using nitrate as a primary nitrogen source will inherently alkalinise the root cytoplasm during the reduction process, as opposed to a plant utilising primarily ammonium. The potential conflict between cytosolic pH and apoplastic pH homeostasis is not discussed. The review should be expanded to discuss issues such as these that might open new research areas.

The manuscript has not been well edited by the corresponding authors. As a minor example, references are cited with surnames only, or with initials and surnames or with full names. It is the responsibility of the authors to ensure consistency in this respect.

---

## [Reviewer Report]

This is an interesting review discussing the signalling role of apoplastic pH variations as well as regulatory mechanisms involved, mainly focussing on the role of the plasma membrane Proton ATPase in these processes.

It describes the current knowledge of the molecular interactions involved. This is a very complex topic, and the reading is very dense, as it is almost entirely made up of a discussion of molecular interactions. Maybe some more words can be dedicated to what is the actual result of all these interactions on plant physiological processes.

Some specific comments:

“73 contexts (Stéger & Palmgren, 2022). Hereafter, we focus part the following sections”

I don’t understand the sentence

“108: (a threonine; T947 in Arabidopsis AUTOINHIBITED H+-ATPase2; AHA2)”

Is this new nomenclature for AHA2?. I was under the belief that AHA2 meant “Arabidopsis H+ ATPase Nr2, otherwise it would be AAHA2”

“118 This functional relationship appears unilateral as phosphorylation of T881 is”

“130 Molecular circuity of auxin-mediated regulation of extracellular pH and growth”

Should this section not be put a bit into historical context referring to the famous “Acid Growth Theory, cell wall loosening etc. The way it is written now one fails to see the physiological importance of all these molecular interactions regulating pHe..

“157 Auxin-induced extracellular alkalinization in roots”

This also is a complicated section, with many detailed but descriptive molecular interactions. It misses physiological explanation/implication. What is the role of auxin in roots, why should it inhibit growth? Why is it doing the opposite in hypocotyls? Maybe it would be easier for the reader to understand if it is first described how root growth is regulated by or responds to external pH, and then explain how auxin and ATPase are involved.

“223 Indeed, in Nicotiana benthamiana several members of a subgroup of intracellular nucleotide-binding leucine rich repeat receptors (NLRs), the plasma

membrane-localized coiled-coil..”

Not clear to me, if they are plasma membrane localized they are not intracellular, or you mean intracellular facing as opposed to extracellular receptors that face the exterior, but are also in the plasma membrane?

“151 prominent example is fusicoccin, a diterpene glucoside produced by the pathogenic fungus Fusicoccum amygdale, which irreversibly stabilizes the interaction between the 14-3-3 and the C-terminal regulatory domain of PM H+-ATPases (Baunsgaard et al., 1998; Jahn et al., 1997) thereby relieving auto-inhibition and leading to constitutive H+ pumping”

Here also I would add the physiological implications: Opening of stomata so the fungus can grow inside. For the other fungi, maybe it is known there also why they inhibit ATPase activity?

“258-262 For instance, the Phytophthora infestans RxLR effector PITG06478 hijacks 14-3-3 proteins to suppress PM H+-ATPase activity (Seo et al., 2023), and the Phytophthora capsici effector CRISIS2 associates with and inhibits PM H+-ATPases to promote disease (Seo et al., 2023). In addition, pathogens and parasite utilize plant mimicking RALF peptides to subvert PM H+-ATPases to their advantage”

So what is the mechanism here to increase disease? Alkaline pHe, but than what happens?

“267-271 while the regulation of PM H+-ATPase has been functionally linked with the regulation of stomata opening, thereby limiting entry of bacteria inside plant tissues (J. Liu et al., 2009; Melotto et al., 2006), it remains however unclear how changes in pHe contribute to immune signaling and ultimately to immunity other cell types.”

Check sentence

“274 How the plant cell senses pH has long been enigmatic (Tsai & Schmidt, 2021).”

Extracellular pH?

Maybe Figures would be clearer if the transporters and receptors are depicted in a more schematic way, with clear arrows indicating antiport, symport etc. Now for instance for SOS1, we see all these regulatory stuff in the cytoplasm, but it is just ugly black spaghetti, not functional for understanding the scheme..

---

## [Editor Report]

Dear authors, your manuscript has now been seen by two reviewers and myself. Although both reviewers find you work interesting, they mention substantial points of criticism that I kindly ask you to address in a major revision. The manuscript would definitely benefit from some careful polishing as it comprises several typos and grammatical errors. Also the quality of the Figures could be substantially improved.

Additionally, I would urge you to correct any scientifically incorrect statements in your manuscript. For instance:

*) l. 59/60 “Prominent examples of molecule/H+ antiporters are […] HKT1 (Schachtman & Schroeder, 1994; Uozumi et al., 2000)…”. The article Schachtman&Schroeder1994 has been corrected by the same laboratory one year later (DOI: 10.1126/science.270.5242.1660). Meanwhile, it is scientific consent that transporters of the HKT-type are not proton/molecule antiporters, but K+/Na+ channels.

*) The notion that AUX1 is a cotransporter with 2H+/IAA- stoichiometry (Graphical Abstract, Figure 1, Figure 2) has no basis. The cited paper assumes such stoichiometry, but does not prove it. Instead, the stoichiometry is still matter of debate and recently a 1H+/IAA stoichiometry has been proposed as the most efficient for thermodynamic reasons (DOI: 10.1111/nph.20120). 

I hope you can address the points of criticism in a major revision. Thank for your contribution to the Research Topic “Quantitative approaches to cellular aspects of plant ion homeostasis”.

Best regards, Ingo

---

## [Reviewer Report]

This is a clearly-written and contemporary account that deals accurately with pH homeostasis in the plant apoplast, particularly as this relates to molecular aspects of development. The review usefully points to areas where further research is needed in the context of relating molecular dynamics to environmental impacts on the plant cell surface in different tissues.

---

## [Editor Report]

Dear authors,

thank you for having revised your manuscript. This clarified the open issues. And thanks again for your contribution to the Research Topic “Quantitative approaches to cellular aspects of plant ion homeostasis”. It is highly appreciated.

Best regards, Ingo